# Philosophia Naturalis Rediviva: Natural Philosophy for the Twenty-First Century

**Bruce J. MacLennan** 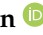

Department of Electrical Engineering and Computer Science, University of Tennessee, Knoxville, TN 37996, USA; maclennan@utk.edu; Tel.: +1-865-974-0994

**Abstract:** A revitalized practice of natural philosophy can help people to live a better life and promote a flourishing ecosystem. Such a philosophy is natural in two senses. First, it is natural by seeking to understand the whole of nature, including mental phenomena. Thus, a comprehensive natural philosophy should address the phenomena of sentience by embracing first- and second-person methods of investigation. Moreover, to expand our understanding of the world, natural philosophy should embrace a full panoply of explanations, similar to Aristotle's four causes. Second, such a philosophy is natural by being grounded in human nature, taking full account of human capacities and limitations. Future natural philosophers should also make use of all human capacities, including emotion and intuition, as well as reason and perception, to investigate nature. Finally, since the majority of our brain's activities are unconscious, natural philosophy should explore the unconscious mind with the aim of deepening our relation with the rest of nature and of enhancing well-being.

**Keywords:** natural philosophy; philosophy of science; Jungian psychology; depth psychology; analytical psychology; phenomenological psychology; evolutionary psychology; active imagination; Aristotle's four causes; aesthetics in science; philosophy as a way of life

---

## 1. Philosophia Naturalis

In the triumphant advance of science, something essential has been lost, but I believe we can recover it by re-examining the idea of natural philosophy. I will begin my exploration with the term philosophia naturalis itself. Originally, philosophia meant, of course, love of wisdom. According to tradition, Pythagoras coined the word because only the gods are truly wise; the best that mortals can do is to desire wisdom and to seek it. This realistic humility is reinforced by the last 2500 years of philosophical and scientific investigations with their continuing revision of previous conclusions. From its beginning, philosophy recognized the limitations of human knowledge.

Traditionally, philosophia was much more than a technical inquiry into the sorts of problems now considered philosophical, and recent commentary has reminded us that ancient philosophy was an all-inclusive way of life [1,2]. Students came to the ancient philosophers and joined their schools in order to live a better life guided by wisdom. The dogmas and technical investigations were important, but primarily as a basis for the art of living well. This goal was also supported by mental and spiritual exercises [1]. We are still concerned with how to live well, and there is growing recognition that philosophy in this broad sense can help us to do so [3–8].

At least from the First Century CE, ancient philosophy was divided into logica (how to understand), physica (understanding of nature, physis) and ethica (character and how to behave) ([9], vol. 1, pp. 158–162). Something like this could be a framework for a future natural philosophy as well. How do we learn and understand? What is the nature of existence? How then do we live? In reconsidering the concept of natural philosophy, I think it is important to take this wider view of philosophy, for we have learned that science and our attitude toward nature have important

consequences for our lives. Therefore, in this paper I will consider a natural philosophy that will help us live better now and help our children to live better in the future. As indicated by the citations, few of the individual ideas are original, but I believe that this comprehensive synthesis into a revitalized natural philosophy is worth defending.

Traditionally, philosophia naturalis could denote the philosophical investigation of the natural world, as opposed to philosophia rationalis (logic), philosophia moralis (ethics) or philosophia divina (theology). But, I think we can construct a more contemporary understanding of natural philosophy by contemplating the adjective naturalis ([10], s.v. naturalis). One meaning of naturalis is "concerning nature" (4b), and therefore, philosophia naturalis has the traditional sense of an inquiry into nature. However, I think it is essential that we understand "nature" in the broadest way, encompassing all the phenomena of our experience, including not just the objective and physical phenomena, but also those considered subjective, personal or mental. A deep understanding of nature, which we require to live wisely, will require exploring outside narrowly empirical and physical phenomena. Later, I will review some means for doing so.

Like the English word "natural", the Latin naturalis has another range of meanings that are especially important for our project. These describe things that have arisen from nature in general or are grounded in it in some way. Such things occur in nature, are part of nature, are produced by natural causes or are determined by natural processes (1, 4a, c, 5a). Then again, naturalis describes characteristics inherent or innate in a thing's nature or typical of it (5e, 7a, 9). From this perspective, philosophia naturalis is naturalized philosophy: philosophy grounded in nature, that is informed by our understanding of nature in general and of human nature in particular. Therefore, in the pursuit of wisdom and knowledge, with the goal of living a better life, we must be cognizant both of nature as a whole and of our own nature. Hence, philosophy grounded in nature depends on philosophy about nature. On the other hand, our investigation of nature depends both on the nature of ourselves as epistemic agents and on the nature of the objects of our investigation, and so philosophy about nature reciprocally depends on philosophy grounded in nature. Therefore, the practice of natural philosophy can be expected to evolve as our consensus understanding of nature and human nature continue to evolve through science and other means of empirical inquiry. Of course, there may be disagreements about the conclusions, as is common in science, but this conception of natural philosophy presumes a fundamental commitment to empirical inquiry, otherwise it cannot, I think, be considered natural philosophy.

As a means of living better, with the goal of human flourishing, natural philosophy should encompass an ethics and morality grounded in human nature and in nature as a whole. Natural morality and ethics and even natural religion and theology are old ideas, perhaps born prematurely, but we know much more now about human evolution, neuropsychology and behavior, and so, the time may be right for their reconsideration and renovation as components of a twenty-first century natural philosophy. This is simply to acknowledge that the characteristics of *Homo sapiens* as a species are relevant to the formulation of ethical norms and to understanding human religious and spiritual beliefs and practices.

## 2. Human Nature

"Know Thyself". A natural philosophy of the sort I am describing depends on an understanding of human nature, which depends on research in psychology, neuroscience, human biology and evolutionary biology. Research in these disciplines is a large and ongoing project; however, there is much that we know, and I will briefly mention some of the characteristics of *Homo sapiens* that are relevant to a future natural philosophy. For the most part, they are uncontroversial and obvious, but we need to call them to our attention.

Certainly, one of the most distinctive characteristics of humans is our ability to learn and adapt; our behavior seems to be more flexible than that of any other animal. This flexibility is a double-edged sword; whereas other species know instinctively how to live authentically as whatever they are,

we have to discover and refine continually what it means to live most fully an authentic human life. For us, living a natural life entails investigating and understanding human nature, so that we can guide our thoughts and behaviors to promote human flourishing. Therefore, in order to serve its traditional function of helping us to live well, natural philosophy should also guide educational philosophy so that we develop and learn well.

Because our learning and adaptation are fueled by knowledge, understanding, insight, wisdom and experience, we are naturally curious. Our natural need to know should be considered a requirement for psychological well-being as essential as are our needs for companionship, love, care, security, stimulation, freedom and peace. Therefore, the quest for wisdom, which is central to natural philosophy, needs no further justification.

Unfortunately, humans have limited cognitive capacity, a characteristic of our species all too familiar to most of us. Our perception, memory and reason are limited in scope and subject to both systematic and random distortion. Our attention is limited and apt to be distracted. Therefore, in our pursuit of wisdom, we need to develop cognitive and other tools to help prevent errors and to detect and correct them when they occur. The methods of logic, mathematics and science are specific examples, but more generally, the social process of scholarship, in which parties with competing interests and agendas critique each other's work, is a means toward eliminating, or at least identifying and mitigating, individual, group and cultural biases. This is perhaps the best we can do.

We are far from understanding the psychological complexity of human nature, which has profound effects on our understanding of ourselves and of the rest of nature, and therefore on our well-being. Humans are sentient beings, by which I mean that they are sensitive to their environments, to their own bodies and to their own interior states, and that this sensitivity is manifest in conscious awareness. Therefore, the natural phenomenon of consciousness, which was ignored by much of Twentieth-Century science, is a fundamental topic for any future natural philosophy. Consciousness has a rich phenomenology including perceptions, thoughts, memories, imagination, inner discourse, feelings, intentions, moods, and much more. In particular, our emotional response, which has often been neglected or even rejected by the philosophy of science, is crucial to our happiness and an important factor in how we reach conclusions, live our lives and interact with other people and the world at large [11,12].

On the other hand, much of what goes on in our brains is unconscious, and so, it is essential that natural philosophers strive to understand these unconscious processes and how they affect philosophers' own psychology, as well as that of other people. We are still, a century or so after the invention of depth psychology, explorers of the complex structure of the unconscious mind, which has enormous effects on all aspects of human life. Central to human nature, it is still poorly understood.

*Homo sapiens* is a social species; we have evolved to survive best in groups, and therefore, social organization is fundamental to our being in the world. Natural philosophy is also a social enterprise, benefiting from the diverse contributions of many people. As a consequence, humans are encultured psychologically and socially through their participation in various communities, and this affects their background assumptions, attitudes, expectations, skills, insights, etc. These cultural characteristics are largely unconscious, slowly acquired and difficult to change. Ultimately, no human activity is culture-free or culture-independent, and it is important that the natural philosopher be aware of this fact (or they will be blindsided by it).

We humans are unique among animals in the complexity and precision of our communication. Language is a cultural artifact that promotes the growth and continuation of culture. It is also an important factor in cognition and even perception, with both positive and negative consequences. Therefore, natural philosophy has to pay special attention to language as an essential characteristic of human nature.

Human beings are embodied, and the significance of that fact is that our brains have evolved to control our bodies in a physical world [13–16]. Our psychological structures are strongly conditioned on embodiment generally and on the specifics of human embodiment. Natural philosophy should not

make the old mistake of treating humans as incorporeal minds contingently and inconsequentially attached to a body. Moreover, as in other animals, human cognition is fundamentally situated, that is rooted in particular situations. Our cognitive faculties are better adapted to concrete physical, social and cultural situations than to abstractions. General insight is harder to achieve and often derived from situated thinking and understanding. Narratives are often more convincing than abstract arguments.

Like other living things, humans have evolved, which means that we have inherited many characteristics that aided our survival in our environment of evolutionary adaptedness, but may be less adaptive in our present, very different environment. It behooves the natural philosopher to be aware of these characteristics of human nature and to take account of them. For example, for 95% of the history of *Home sapiens*, we survived as hunter-gatherers in small groups of related individuals ([17], pp. 87–88). That is our environment of evolutionary adaptedness, but that does not imply that we should live as paleolithic foragers or that we should accept today the behaviors that were adaptive then. Important characteristics of human nature are that we learn, adapt, cooperate and pass on our collective experience through culture, which itself evolves. Therefore, for us to live now as paleolithic foragers would be profoundly unnatural, contrary to authentic human nature. Indeed, I expect that the natural philosophy of the future will be an important contribution to the evolution of culture.

Finally, human beings are mortal, and so, the continuation of humanity depends on reproduction and the ability of our offspring to survive and flourish. Therefore, a natural philosophy should be forward-looking and focus on future generations. Like other animals, humans must act purposefully for their own survival and to ensure the survival of the species. Also in common with other species, human survival depends on the health of the ecosystem, and beyond mere survival, the well-being of humanity depends on a flourishing ecosystem. Indeed, biophilia is an evolved appreciation for a healthy environment, which is part of human nature and fundamental to our well-being [18].

As a future natural philosophy should be informed by human nature, so also it should start from the fact that humans are a part of nature. The global ecosystem is an integrated and organized whole, and as such, we may ask what role humans play in it [19]. On the one hand, we now understand that humans have a greater effect on the environment than do other species. On the other, humans have unique capacities for understanding and influencing nature, and we may use them to enhance the survival and flourishing of the global ecosystem, on which we all depend. Just as we individually use our sense organs and minds to better adapt to our environments, so humankind can serve as an organ for the adaptation of the ecosystem as a whole. That is, we can make ourselves part of the global ecosystem feedback loop and work to enhance its health rather than to harm it. However, if humanity is going to fulfill this function well, it will need to strive to understand the whole of nature, and we may consider what that entails.

## 3. Three Perspectives

One of the facts about nature that a complete natural philosophy must accommodate is the existence of sentient beings, including of course human beings, but also many—if not all—other animal species. Sentient beings have two aspects: an exterior as a physical object and an interior as a consciously-aware subject. These aspects necessitate two perspectives, commonly termed third-person and first-person.

From a third-person perspective, a sentient subject or observer seeks to understand a physical object in terms of its external behavior and physical structure, that is by addressing its non-sentient aspects. This is the perspective of the physical sciences and of behaviorist psychology, but also of much cognitive science and neuroscience, which treat cognition as physico-chemical information processing and control.

Certain natural phenomena cannot be observed directly from a third-person perspective, and these include the subjective structure of sentience and phenomenal consciousness. For understanding these phenomena, first-person methods have been developed, as in phenomenological psychology and experimental phenomenology [20–22]. While third-person investigations can address these phenomena

indirectly, the most fundamental problems (such as the Hard Problem of consciousness [23]) cannot be solved without evidence available only from a first-person perspective.

First-person investigations are more difficult than third-person research for several reasons. First of all, first-person methods have not been so extensively developed and refined as third-person techniques. Second, due to the private nature of first-person investigations, there is a greater danger of personal biases, presuppositions and other subjective factors affecting observation. Third, and most importantly, this privateness makes public observation in principle impossible. Nevertheless, publicly-validated understanding can emerge in a community of investigators through shared practices of introspection and experimentation [24] (we already find this shared understanding in well-established contemplative and meditative communities). Moreover, first- and third-person approaches can be combined, as in neurophenomenology [24–26].

The first-/third-person grammatical analogy encourages us to consider whether there is also a second-person perspective, and I believe that there is and that it will become an important part of natural philosophy [19]. The first-person perspective has a sentient subject striving to understand his/her own subjectivity, that is to understand his/her interiority from the inside; and the third-person perspective has a sentient subject striving to understand an object from an external standpoint, that is qua non-sentient thing. The second-person perspective, in contrast, has two or more sentient beings striving to understand one another qua sentient beings, that is each understanding their own interiority in relation to the interiorities of the others. It is a cooperative activity of mutual growth.

I believe that the second-person perspective is fundamental to phenomenology, for we are social beings relating to other sentient beings before we ever undertake first-person phenomenology, which has a solipsistic orientation. The first-person perspective is a bracketing of experience from everyday second- and third-person relationships. If we are to obey Husserl's "Back to the phenomena!", then we must acknowledge the second-person perspective.

An everyday example of the second-person perspective is the mutual understanding that develops between close friends, lovers and family members. Good examples of systematic formal second-person investigations might be the relation of the Jungian analyst and analysand and other psychotherapeutic or long-term counseling relationships. Second-person understanding can also develop between humans and non-human sentient beings. A familiar example is the understanding that arises between people and their companion animals. There is a partial recognition of the second-person relationship in contemporary rules and guidelines in human subjects research and in animal research, which acknowledge the objects of the research as sentient beings whose experiences, sensibilities and autonomy should be considered.

In summary, natural philosophy should investigate nature from first-, second- and third-person perspectives, which may be described as intrasubjective, intersubjective and objective (more properly, subjective-objective) investigations. These three perspectives are necessary for complete understanding in a world in which there are sentient beings.

## 4. Four Explanations

Understanding the why of things is central to natural philosophy, but there are several sorts of answers to why questions. In any given context, some kinds of answers, or explanations, may be more or less informative—more or less able to improve our understanding—than others. However, the contraction of natural philosophy that accompanied the expansion of modern science in the Sixteenth through Eighteenth Centuries led to a corresponding contraction in the notion of causality. The newly dominant mechanical philosophy explained all causation in terms of efficient causation, which is still the common scientific approach. In the broader context of natural philosophy, the efficient cause of an event is not always the most informative explanation. Therefore, as a first step toward a broader understanding, we can reconsider Aristotle's analysis of answers to why questions (Aris., *Phys*. II 194b–195a, *Met*. 983a–b, 1013a–1014a). These are commonly known as Aristotle's four causes, but that terminology can be misleading due to the limited notion of causality typical of

contemporary science. Therefore, I prefer to call them the four whys or, compromising with tradition, the four (be)causes. Nevertheless, it is important to understand that the four whys are not a theory of causation, but a taxonomy of explanation. A brief review follows, which puts them in the context of future natural philosophy.

One fundamental kind of explanation can be termed the "what" (Greek, *to ti esti*), which answers the question "What is it?" Traditionally, this is called the formal cause (*causa formalis*) because the answer refers to the form, class or category to which something belongs (Grk., *eidos*). Why does this thing have feathers? Because it is a bird, and birds have feathers. Why did this animal pounce on the bird? Because it is a cat, and cats prey on birds. Why did this tissue contract? Because it is a muscle.

In the context of formal causation, "formal" refers to the Platonic forms or ideas (Grk., *eidos, idea*), and so, formal causes also include mathematical explanations, which have been essential in science since Galileo's time. Why do these two electrons repel each other with such and such a force? Because electrons are charged objects, which obey Coulomb's law, and so, the force is proportional to the product of their charges and inversely proportional to the square of their distance. Why is $D_x(x^2 + \sin x) = D_x x^2 + D_x \sin x$? Because differentiation is a linear operator.

A second sort of explanation is the "from what" (Grk., *to ex hou*), which answers a why question in terms of the material from which something is formed; this is the material cause (*causa materialis*). In this context, "material" (Grk., *hulê*) is not limited to the sort of physical matter from which something is made, but is relative to a thing's form. That is, the thing we are seeking to explain is analyzed in terms of some form imposed on an underlying substrate, its "matter". The formal (be)cause refers to a specific abstract class, category or form; the material (be)cause refers to the generic unformed stuff from which the thing is formed. Why did the house burn down? Because it was made of wood. Why did the cat fall? Because it is made of flesh and blood (which have mass, etc.). Why did the muscle contract? Because it is composed of thousands of muscle fibers, each of which can contract.

Form and matter are often relative terms, for the formed matter at one level becomes the generic substrate for higher levels of formation. A statue (to use an old example) has many properties, some better explained by its form (it is a statue of Apollo), others better explained by its material (bronze). But, the bronze metal is itself formed matter, for it is an alloy of copper and tin in a particular proportion, and copper and tin are themselves structures of more elementary matter (protons, neutrons, electrons) with a certain crystal structure, and so forth. At a higher level, statues may be the matter of a museum exhibition.

A third sort of explanation is the "by what" (Grk., *to hupo tinos*) or efficient cause (*causa efficiens*), which is the sort of explanation privileged by contemporary science. Aristotle tells us that this answer to a why question explains a change in terms of what initiated the change, maintains it or brought it to completion. Thus, it explains a change, typically in terms of another change (Grk., *kinoun*), either antecedent, concurrent or terminating. Why did the ball fly over the net? Because it was struck by the racket. Why did the cat pounce? Because it saw a bird in range. Why did the muscle contract? Because it was stimulated by motoneurons.

The most controversial kind of explanation, from a contemporary perspective, is the "for sake of what" (Grk., *to hou heneka*), or final cause (*causa finalis*), which explains something in terms of its end or purpose (Grk. *telos*). Why does the heart beat? To pump the blood. Why are there antibiotics? To fight infection. Why did the cat pounce on the bird? In order to eat it. Why did this muscle contract? To extend the cat's legs so it could pounce.

Indeed, many things in nature—especially in living nature—exhibit teleonomic behavior; that is, they behave in such a way that they fulfill purposes or achieve relevant ends ([27], pp. 9–20). Contemporary science prefers to explain them in terms of antecedent efficient causes (e.g., natural selection and myriad contingencies), but especially in biological and technological contexts, final (be)causes are often more explanatory. In fact, contemporary evolutionary theory explains how teleonomic processes arise in the natural world, and all four be(causes) are essential to explanation in

modern evolutionary biology [28,29]; as Pigliucci observes,"Darwin made it possible to put all four Aristotelian causes into science" [30].

In summary, all four whys or (be)causes are necessary for a complete understanding of anything. The material and formal explanations say what a thing is in generic and specific terms; the efficient cause addresses the motive forces of its change; and the final explanation identifies the purpose or function of the change. Certainly, for any particular thing and for any particular purpose, some explanations will be more relevant, some less. But, in order to achieve better understanding and greater wisdom, natural philosophy should be open to them all.

We should not assume, however, that Aristotle said the first and last words on the categories of explanation. Certainly, natural philosophy should be open to new forms of questions and answers that better enable us to understand nature in all its manifestations. Nevertheless, there is something fundamental about Aristotle's framework, which looks for explanations in the past (efficient), in the future (final) and in present nature combining general law (formal) and particular substance (material).

## 5. Philosophical Practice

How should we practice natural philosophy? I have argued that it is a philosophy grounded in nature and, in particular, in human nature. Therefore, we must take account of all of human nature, and not ignore some aspects of it or attempt to wish them out of existence. Rather, we should consider every aspect of human nature as a means of achieving greater understanding with wisdom as our ultimate goal. As our understanding of human nature extends and deepens, so also will our understanding of how to pursue wisdom.

### 5.1. Four Functions

The characteristic of human nature most immediately apparent to us is our conscious mind, and therefore, we may begin with its faculties and how they may be applied to natural philosophy. C. G. Jung identified four orienting and adaptive functions of the conscious mind: thinking, sensation, feeling and intuition ([31], CW 6, ¶¶ 7, 983–985).[1] Although Jung's taxonomy might not be exhaustive, it has stood the test of time, and I will use it here. Sensation refers to conscious perception of the external world, and thinking refers to our ability to reflect on our mental content, especially by discursive and rational means. Sensation and thinking have been the faculties most obviously applied in science for the last several centuries and broadly align with empirical and theoretical investigation. The former is more extroverted in its orientation, the latter more introverted. Less obviously useful in science are the other two faculties—feeling and intuition—but we have testimony to their importance from some of the greatest scientists.

The feeling function provides an assessment of something that has the immediacy of sensation (indeed, its biological function is to provide an actionable assessment when more thorough, but slower thinking is not practical). Feeling has a valence: positive or negative, attraction or avoidance, good or bad, but also other dimensions that are difficult to characterize ([32], p. 90).

Many scientists have commented on the importance of aesthetic considerations in guiding their own work [33], even sometimes in opposition to empirical evidence, with eventual vindication of the more aesthetic theory ([34], pp. 65–66). There does not seem to be any a priori reason to prefer the more aesthetic theory, unless one takes the Platonic view that Truth, Beauty and the Good are aspects of the ultimate principle of existence, but aesthetics is often a reliable guide. Perhaps it is simply that our brains work better on aesthetically-appealing material. We are more likely to dwell on and even contemplate the beautiful, and thus find aesthetic theories more fruitful. Especially in mathematics and mathematical sciences, beauty is associated with order, symmetry and harmony,

---

1　Jung's work is cited by paragraph number (¶) and volume in his Collected Works (CW).

which facilitate thinking. The pursuit of truth may be guided by aesthetics more efficiently than by slower discursive reasoning.

In any case, the role of aesthetics in understanding should be a topic of investigation for natural philosophers; it is a characteristic of human nature that needs to be better understood. Aesthetic cultivation is an implicit part of the training of most mathematicians and scientists, but it could be taught more explicitly. Now, it is learned through apprenticeship and individual discovery, but we could have courses intended to cultivate the natural philosopher's aesthetic judgment. Different cultures and even different scientific and philosophical communities have different aesthetic values, and studying this diversity will expand the aesthetic horizons of natural philosophers.

Aesthetics is just one aspect of the feeling function, which has components that are both innate and learned. Our emotional responses have evolved over millions of years to make rapid—and on average, reliable—evaluations in our environment of evolutionary adaptedness [35]. These responses are not necessarily adaptive in our contemporary, very different environment, and so, we regulate and modify our emotional responses through cultural conditioning and learning. Nevertheless, as perceptual organs, our emotions give us valuable information, especially about people and, to a lesser extent, other animals. Therefore, they are especially important in second-person investigations.

As perception makes use of sense organs, which process sensory information unconsciously before it becomes present in conscious perception, the feeling function is also embodied in brain structures such as the amygdala and other parts of the limbic system, with unconscious effects on the physiological state. Before an emotional response rises to the level of consciousness and becomes present to the feeling function, it has already had physiological effects, such as activation of the sympathetic nervous system, hormone secretion (e.g., adrenalin) and alteration of breathing and heart rate. Our conscious awareness of such effects is essential to the phenomenology of the feeling function ([36], ch. 7) ([37], ch. 9). Therefore, cultivation of the feeling function involves greater awareness of the somatic correlates of emotion. Where am I feeling this? In my gut? In my heart? In my breathing?

From ancient times up to the present day, science and to a large extent also philosophy have been prejudiced against the feeling function, but it is essential. Indeed, people with a pathological absence of feeling cannot make decisions effectively ([36], p. 67). Nevertheless, the emotional faculties, which evolved in a very different environment from modern civilization and often develop in the individual without much conscious reflection, cannot be relied upon blindly. Like our sense organs and indeed our thinking, our feelings can be misleading. Therefore, it is important to treat our emotional responses critically and to cultivate them to respond more appropriately in contemporary and future society.

In natural philosophy, the feelings are not sufficient on their own (nor are the other three functions), but they are often necessary for complete understanding. In particular, when properly cultivated, they may give us an early assessment of an idea and help us to decide whether it is worth pursuing by means of perception and thought. Moreover, after perception and thought have done their job, the feelings can help us evaluate the quality of the result.

This brings us to the fourth function, intuition, which is perhaps the least familiar. Jung compares intuition with the other functions as follows: "The essential function of sensation is to establish that something exists, thinking tells us what it means, feeling what its value is, and intuition surmises whence it comes and whither it goes" ([31], CW 6, ¶983). He also defines intuition as "perception by way of the unconscious, or perception of unconscious contents" ([31], CW 6, ¶899); it "should enable us to divine the hidden possibilities in the background, since these too belong to the complete picture of a given situation" ([31], CW 6, ¶900). Intuition is the faculty that brings new possibilities into conscious awareness; it is the fundamental organ of creativity. From it arise novel ideas, hypotheses, images and visions, which then may be subjected to critical evaluation by the thinking, sensation and feeling functions.

To make full, conscious use of their intuitive faculties, natural philosophers need to learn and practice techniques for bringing unconscious content and processes into conscious awareness. These techniques include active imagination and attention to dreams [38]. This may seem far outside the

bounds of traditional science and philosophy, but there are many examples from history of the creative potential of intuition. Perhaps the most familiar is Kekulé's discovery of the benzine ring; "Let us learn to dream", he advised, "then perhaps we shall find the truth" [39,40].

A fundamental conclusion of the Jungian psychological typology is that most people have one dominant function, which is the principal mode of their conscious engagement with the world. It is their most differentiated function, the most fully developed and precise and the one they habitually use. The opposite function (thinking and feeling are opposites, as are sensation and intuition), which is called the inferior function, then is the least differentiated and developed and may be quite primitive in its functioning, which is often largely unconscious ([41], pp. 10–18). A person is least likely to use his/her inferior function, and when they do, he/she often does not use it effectively, due to its underdevelopment. The remaining two functions are called secondary or auxiliary and have intermediate degrees of differentiation and use.

Thinking is the dominant function for most scientists and philosophers, with sensation an auxiliary function, especially for empiricists. Feeling and intuition are usually the less developed functions. All four functions, however, are human faculties for conscious adaptation and orientation in the world, and Jung informs us, "For complete orientation all four functions should contribute equally" ([31], CW 6, ¶900). In this way, we have complementary perspectives on any phenomenon, essentially seeing it from all four sides. Developing the secondary and inferior functions is part of the psychological process of individuation, of becoming psychologically whole and undivided (Latin, *individuus*), which is the goal of Jungian analysis [42]. An especially challenging, early phase of individuation is familiarization with and recruitment of the Shadow complex, which incorporates consciously-rejected characteristics such as the inferior function ([42], pp. 38–42, [43]). The engagement with the Shadow integrates these unconscious characteristics into consciousness. So, a thinking-dominant scientist would need to become more consciously aware of his/her largely unconscious feeling function, and to work with it so that it is a more adaptive, differentiated and useful faculty. A goal for the education of future natural philosophers should be the cultivation of all four functions, so that they have all their conscious faculties available for understanding the world and living better in it.

*5.2. Unconscious Faculties*

Much of what takes place in our brains is unconscious, and it behooves us as natural philosophers to understand our unconscious faculties, both from the third-person perspectives of neuroscience and behavioral psychology and from the first-person perspectives of phenomenology and analytical psychology. In fact, all the conscious faculties have roots in the unconscious. We have seen that early phases of emotional processing are unconscious, and in perception, both early stages (e.g., pattern recognition) and top-down processes (e.g., "seeing as") are unconscious. So, also the possibilities presented to the intuition arise from the unconscious. Even the thinking function leans heavily on largely unconscious processes, such as categorization and concept formation, memory and language.

Like other animals, humans have evolved behavioral adaptations ("instincts") that are characteristic of *Homo sapiens* (i.e., phylogenetic). They are encoded in the human genome and the subject of ongoing research in evolutionary psychology [44]. These innate adaptations lie deep in our nature and define the phylogenetic core of our unconscious minds; Jung called it the objective psyche ([31], CW 7, ¶103n, [45], p. 65, [46]) and the collective unconscious because it is common to all people ([31], CW 8, ¶270). The particular instincts structure our perception, affect, motivation and behavior to achieve biological ends (e.g., reproduction, child rearing, cooperation, social hierarchy, protection). Instincts can be studied from a third-person perspective by observing behavioral regularities characteristic of a species. They regulate behavior, however, by means of their effect on conscious and unconscious processes in the animal's brain. Experienced from a first-person perspective, these structures are the archetypes of the collective unconscious, which Jung described as "active living dispositions, ideas in the Platonic sense" ([31], CW 8, ¶154).

The archetypes are often misunderstood as innate images; indeed, this was Jung's initial understanding of them ([31], CW 8, ¶435), but in his later work, he stressed that they are not images, but innate regulators of psychological processes ([31], CW 9i, ¶155). Therefore, the archetypes are less like static patterns and more like programs or control systems that regulate behavior by means of the nervous system; they are dynamic psychological forms, that is structured regulators of behavior and experience.

As dynamic psychological forms, the archetypes shape the particular "matter" of our behavior and experience ([31], CW 9i, ¶155). Together, that is, they are (partial) formal and material explanations of our thought, feelings and action. The final explanation lies in the biological ends served by the archetypes. The efficient explanation is the releasing stimulus that has activated the archetype, that is engaged a cognitive-behavioral regulatory mechanism ([45], pp. 64–65). Of course, I am not claiming that every human thought, feeling or action can be explained by the archetypes, but as evolved characteristics of our species, understanding them is essential to any natural philosophy.

There is more to the unconscious mind than the collective unconscious, for each of us also has a personal unconscious, which is ontogenetic rather than phylogenetic ([42], p. 150n13). It develops in each of us as individuals in particular families, communities, groups and cultures. The personal unconscious is largely an adaptation of phylogenetic archetypes to the particularities of an individual's life. This adaptation takes the form of unconscious complexes, each developing around an archetypal core. In common usage, the word "complex" has a negative connotation, but in the context of analytical psychology, complexes are normal components of the unconscious ([41], pp. 36–39). They are what makes the human instincts flexible and subject to individual, social and cultural modification. Nevertheless, because complexes develop unconsciously through a person's life experiences, they can become maladaptive. Therefore, an important goal of Jungian psychoanalysis is to bring the complexes into conscious awareness, so that the analysand can engage with them and so that their unconscious effect is mitigated (this has been discussed above in Section 5.1 for the specific case of the inferior function and the Shadow complex).

### 5.3. Active Imagination

Archetypes and complexes are unconscious and therefore not directly observable, even by first-person methods. But, like other theoretical entities in science, they may be investigated through their observable effects, which allows hypotheses about them to be confirmed or refuted. When archetypes and complexes are activated or engaged, they have effects on experience, and we can come to understand them through these experiences. In particular, Jung observed that archetypes and complexes often behave as autonomous subpersonalities with their own purposes (deriving from their biological function) ([31], CW 8, ¶253). Their inner workings and motivations are not directly accessible to consciousness (for they are opaque to us, like the phenomenological interiors of other sentient beings), but we can engage them in a second-person investigation. More concretely, the conscious ego and an unconscious complex/archetype can engage in a dialogue directed toward mutual understanding: the ego of the complex's goals and needs, the complex of the ego's individual life and needs in the here and now, with the goal of a mutual accommodation ([38], pp. 179–188). In this way, a cooperative relationship is established, rather than a situation in which the components of the psyche work at cross-purposes.

Active imagination is the name given in analytical psychology to the principal technique for achieving this accommodation [38,47]. The practitioner consciously interacts in his/her imagination with a personification of an activated complex or archetype. The dialogue (and, indeed, negotiation) is only partly under the control of the ego, for the activated unconscious personality is governed by its own autonomous structure. Active imagination depends on a complex or archetype being activated in the unconscious. Sometimes, this happens spontaneously, for example when a person has had an especially impressive dream. In this case, a significant person, animal or object from the dream can be used as an imaginative stimulus to reactivate the relevant complex or archetype. In other cases,

a person may invite a personification of a particular affect or condition (such as a mood or illness) that is intervening in their life.

More generally, complexes and archetypes are activated by symbols, which acquire their numinous character because they are the releasing stimuli of these deep psychological structures ([48], pp. 12–44). Symbols seem significant because they are significant, signifying situations in which some associated archetype or its derivative complexes should be engaged. The symbol's numinosity is a conscious manifestation of the activation of an archetype or complex in the unconscious psyche. Some symbols are apparently innate, wired into our psyches through hundreds of thousands of years of evolution. Others are more particular—cultural or even individual—and become associated with archetypes by means of their mediating complexes. These are not new ideas. In particular, Neoplatonic theurgists used symbols (sumbola) and signs (sunthêmata) to engage with archetypal figures and complexes, which they understood as gods and daimones (mediating spirits) [49,50]. They accomplished this through ritual, which may be defined as "symbolic behavior, consciously performed" ([38], p. 102).

Active imagination and similar techniques are not just for dealing with psychological problems, but can be valuable philosophical tools, as the Neoplatonists knew. The unconscious mind has long been recognized as a source of creativity [51–55], but creators have had to wait for the unconscious to offer on its own terms new possibilities to intuition. Active imagination enables conscious engagement with the wellsprings of creativity in order to discover new possibilities, which then may be evaluated according to the criteria of the other three functions (thinking, sensation, feeling). This too is an old idea, and ancient poets' invocation of the Muses as a source of inspiration could be more than a literary convention. By symbolic actions, a person can activate the relevant archetypes and complexes and seek inspiration from them. So, also the natural philosopher may seek insights about himself/herself and the rest of nature. Archetypes, complexes and symbols are characteristics of human nature that we, as natural philosophers, need to understand better, both to understand ourselves more deeply, but also to understand their effect on our understanding of other things.

### 5.4. Natural Philosophy of Mathematics

The natural philosophy of mathematics illustrates the importance of the unconscious. It has been argued convincingly that the only viable philosophies of mathematics are fictionalism and full-blooded Platonism, but that there is no fact of the matter to decide between them ([56], pp. 4–5 and ch. 8). In both cases, mathematical objects exist (roughly speaking) if there is a consistent theory about them. This seems to be an impoverished view of mathematical reality compared to the experiences of many mathematicians and scientists. The gap arises from the fact that in philosophy, mathematical objects are treated purely formally; in the case of the natural numbers, they are understood as pure quantities. In fact, Wolfgang Pauli argued that contemporary work in the foundations of mathematics had failed because its formal approach "was one-sided and divorced from nature" ([57], p. 64).

However, if we take the perspective of natural philosophy, we see that the natural numbers also have a ground in human nature; for humans (like some other animals) exhibit innate numerosity [58,59], that is the ability to directly perceive small numbers. There are regions of our brains that respond to the numbers of things independently of their other properties (size, arrangement, density, etc.) [60]. Like our phylogenetic capacity to see form, color and arrangement, we have a phylogenetic capacity to see number (up to about seven). Innate numerosity together with innate symmetry perception [61] imply that these mathematical concepts are not arbitrary constructions for us; they are implicit in the human genome and have been with us for a very long time. They are archetypal.

Therefore, certain small numbers have inherent qualitative aspects in addition to their more familiar quantitative aspects. This has been recognized by mathematicians such as Henri Poincaré, who said, "Every whole number is detached from the others, it possesses its own individuality, so to speak" ([62], p. 60) and by physicists such as Pauli, who said the archetype concept should include "the continuous series of whole numbers in arithmetic, and that of the continuum in geometry" ([62], p. 18n10). In psychological terms, the first few numbers are individually archetypal, and they have

psychological potency like the other archetypes. Indeed, "Jung devoted practically the whole of his life's work to demonstrating the vast psychological significance of the number four", according to his colleague Marie-Louise von Franz ([62], p. 115). Jung remarked that number "may well be the most primitive element of order in the human mind" ([31], CW 8, ¶870).

A complete natural philosophy of mathematics should acknowledge and incorporate the archetypal character of small numbers and geometric objects, for they are grounded in human nature. The more abstract qualities of symmetry are of course fundamental to the aesthetics of mathematics and physics. Pauli emphasized the importance of the archetypes in natural science; he concluded, "the archetypes thus function as the sought-for bridge between the sense perceptions and the ideas and are, accordingly, a necessary presupposition even for evolving a scientific theory of nature" ([63], p. 221).

## 6. The Reanimation of Nature

The last natural philosopher, in the sense presented here, was perhaps Johann Wolfgang von Goethe, and the present proposal could be viewed as an attempt to pick up where he left off, but in the context of a Twenty-First Century understanding of nature. His approach, which he called a "delicate empiricism" (zartre Empirie) involved an empathetic identification with the object of study, so that exterior perception and interior cognition move in tandem harmony; he said, "my perception itself is a thinking, and my thinking a perception" ([64], p. 39). His method was holistic and participatory ([63], pp. 146, 258, [65], pp. 3–26, 49–76, 321–330, [66], pp. 12, 22, 28, 41, 48). Archetypal patterns provide the unifying bridge between external forms and processes and an empathetic participation in them. Pauli agrees that understanding, and in fact the joy of understanding, arises from "a correspondence, a 'matching' of inner images pre-existent in the human psyche with external objects and their behavior" ([63], p. 221) (I have discussed Goethe's natural philosophy more elsewhere [19,50]).

Natural philosophers, as described here, will interact with nature in a way that is more holistic, participatory and sensitive than is the norm in science now. They will engage all four of their conscious functions with a goal of understanding phenomena that are intrasubjective, intersubjective and extrasubjective (objective). They will cultivate relationships with the complexes and archetypal structures of the unconscious mind and will be aware of the projection of these structures outside themselves. In this way, they will attune their psyches to nature to achieve a fuller, more comprehensive understanding and to gain the wisdom that is built on it. This resonance between inner and outer form and process will allow them to experience nature as animate, which can be grasped and appreciated not just intellectually, but also sensually, emotionally and intuitively [67]. I anticipate this will change the relationship of humanity to the nature of which it is part to the benefit of both.

## 7. Conclusions

In this paper, I have presented what I believe natural philosophy should be in the Twenty-First Century. First of all, it should be a philosophy in the ancient sense, that is a way of living better, of human flourishing grounded in the pursuit of wisdom.

Second, it should be a natural philosophy in that it finds wisdom by seeking to understand all of nature, which includes all the experiences we have, both individual and collective. In particular, in addition to understanding objects qua objects from an exterior, third-person perspective, it seeks to understand experience from an interior, first-person perspective, and it seeks mutual understanding among sentient beings through a second-person perspective. No phenomenon should be outside the scope of a future natural philosophy. Moreover, natural philosophers understand that phenomena have many explanations, and the most informative explanation of a phenomenon may lie in antecedent changes, but also in specific structure or organization, in generic constituents or in the purpose or function of the phenomenon, as well as that full understanding often depends on all of these.

Third, it should be a natural philosophy in that it is grounded in our nature as human beings. This means that it is a natural philosophy developed in an ever-improving awareness and

understanding of human nature, in particular of human capacities and limitations. Therefore, future natural philosophers should use all their faculties in the pursuit of wisdom: conscious and unconscious, individual and collective. We are situated, embodied living agents with capacities for thinking, sensation, feeling and intuition, all of which are informative. Part of natural philosophy as a way of life, then, is the cultivation of these capacities so that they function more effectively. In particular, feeling and intuition need more attention than usually granted by science. To this end, particular contemplative practices and exercises will be helpful.

As human animals who are part of nature, we may use our innate capacities to enter into a comprehensive, empathetic understanding of nature, which is intellectually and emotionally satisfying and which leads to humanity better fulfilling its function as an organ of nature. A better understanding of nature, including human nature, will show us how to fulfill our role better, and a better understanding of our own nature will enable us to understand better nature as a whole. It goes without saying that most if not all of these ideas have been proposed before, but the goal of a philosophia naturalis rediviva must take the best of the past while establishing a new foundation on which to erect a revitalized structure.

**Funding:** This research received no external funding.

**Conflicts of Interest:** The author declares no conflict of interest.

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
