# Peer review of "Philosophia Naturalis Rediviva: Natural Philosophy for the Twenty-First Century"

_philosophies, doi:10.3390/philosophies3040038_

Round 1

Reviewer 1 Report

Q: Since you bring in human properties, how would you deal with ideological differences? Your perspective implies an ideology. If you state your background explicitly you will likely drive away thinkers from other backgrounds (Marxists, Muslims, etc.). 

Second Person discussion interesting & thought provoking

Here he asserts a "goal" of natural philosophy -- 'wisdom'. This goes beyond the goal of understanding (which my contribution to this collection could be said to hope to elicit). This “wisdom” as the goal of natural philosophy results from including human nature in its study.

While I try in my paper to clarify what is known, he tackles mostly the knower

(Ref: ‘The Knower and the Known’  Marjorie Grene book)

The section on archetypes leaves me somewhat befuddled. Especially we need some kind of definition of archetype. Are they cultural personality attractors?  Can some clear comparison with the objects that are the subjects of science be made?  Are they more general or less general?  How about: 

                        {natural science {anthropology {psychology}}}? 

You suggest, seemingly in contrast with archetypes, that numerosity appears to be genetically held rather than culturally imposed. But you go on to write that the first few numbers are archetypical. Thus, might some archetypes be genetically predisposed?

Reviewer 2 Report

The proposal has the elements of a theory of everything, where philosophy=philosophy of nature; existence=natural existence; theory=Weltanshauung, where some obsolete terms are used (archetype) instead of nowadays ones (pattern), where philosophy of nature=Gaia's philosophy.

For a Journal with the theme "Philosophies" it is ok as it illustrate a sincere love for philosophy and love for the future of humankind.

However, stricto sensu, this is a modern, not contemporary plead for philosophy, only.

Please see detailed comments in the attachment.

Reviewer 3 Report

The paper is well written, but both tone and content leave me perplexed.

I find it difficult to classify the paper; more than philosophy, it seems to me concerned with psychology, psychoanalysis, education.

Rather than proposing a new natural philosophy,  the author proposes the abolition of specializations in philosophy. This is my impression. 

Personally, I believe that all the characteristics desired by the author for the new natural philosophy are contained in the various branches of modern philosophy and science.

Referring to the four causes of Aristotle after the sophisticated theories of causation of the nineteenth and twentieth centuries does not seem tenable.

Perhaps the most centered criticism is that of mathematics, which can be extended to all the modern sciences. Here, greater attention to philosophy and a partial recovery of classical natural philosophy would be desirable.

Concluding according to me, the work cannot be published in the present form

Round 2

Reviewer 3 Report

I am not fully satisfied but the paper has improved